# Microenvironmental IL-6 inhibits anti-cancer immune responses generated by cytotoxic chemotherapy

Eric H. Bent[1,2,3], Luis R. Millán-Barea [1,2,3], Iris Zhuang[1,2], Daniel R. Goulet[1,2], Julia Fröse[1,2] & Michael T. Hemann [1,2✉]

Cytotoxic chemotherapeutics primarily function through DNA damage-induced tumor cell apoptosis, although the inflammation provoked by these agents can stimulate anti-cancer immune responses. The mechanisms that control these distinct effects and limit immunogenic responses to DNA-damage mediated cell death in vivo are currently unclear. Using a mouse model of BCR-ABL[+] B-cell acute lymphoblastic leukemia, we show that chemotherapy-induced anti-cancer immunity is suppressed by the tumor microenvironment through production of the cytokine IL-6. The chemotherapeutic doxorubicin is curative in IL-6-deficient mice through the induction of CD8[+] T-cell-mediated anti-cancer responses, while moderately extending lifespan in wild type tumor-bearing mice. We also show that IL-6 suppresses the effectiveness of immune-checkpoint inhibition with anti-PD-L1 blockade. Our results suggest that IL-6 is a key regulator of anti-cancer immune responses induced by genotoxic stress and that its inhibition can switch cancer cell clearance from primarily apoptotic to immunogenic, promoting and maintaining durable anti-tumor immune responses.

[1] The David H. Koch Institute for Integrative Cancer Research, Cambridge, MA 02142, USA. [2] Department of Biology, Massachusetts Institute of Technology, Cambridge, MA 02142, USA. [3] These authors contributed equally: Eric H. Bent, Luis R. Millán-Barea. ✉email: hemann@mit.edu

Most conventional chemotherapeutics exert their cytotoxic mechanism of action by interfering with diverse proteins that affect DNA synthesis and replication. These cellular disruptions lead to the induction of genotoxic stress which results in DNA damage and ultimately in cell death[1]. Most cancers are initially treated with conventional chemotherapeutics, but complete tumor eradication is difficult to achieve with either targeted or cytotoxic agents. Persistent disease, frequently termed minimal residual disease, fuels eventual tumor relapse and treatment failure in many patients, underscoring a need to find ways to enhance the long-term efficacy of our front-line arsenal of cancer therapeutics.

Some of the most widely used chemotherapeutics, such as doxorubicin, have been suggested to induce anti-tumor immunity through the stimulation of immunogenic cell death (ICD)[2]. The generation of anti-cancer immunity is a promising approach to target residual disease in cancer, and can result in durable tumor responses[3,4]. Lasting anti-cancer immune responses require both antigen recognition and adjuvant signals, such as those that result from cell stress or death[5,6]. Immune-stimulating chemotherapies incite the release of pro-inflammatory signals, including damage-associated molecular patterns (DAMPs), that indicate danger and act as immunologic adjuvants, provoking anti-tumor immunity. However, even in settings where tumor antigens are present, cytotoxic chemotherapy rarely generates durable anti-cancer immune responses. This suggests that any immune stimulus from genotoxic therapy is insufficient or ultimately suppressed. The mechanisms by which this occurs are not well understood.

BCR-ABL[+] B-cell acute lymphoblastic leukemia (B-ALL) is a treatment-refractory subtype of B-ALL with a ~50% 3-year overall survival following the use of cytotoxic chemotherapeutics in combination with targeted BCR-ABL kinase inhibitors[7,8]. Most chemotherapy regimens for ALL include the anthracycline doxorubicin, which can promote ICD and has the potential to induce anti-tumor immunity[2,7,8]. However, patients with BCR-ABL[+] B-ALL rarely experience immune-mediated cures after doxorubicin therapy. Immune evasion is a hallmark of cancer development[9,10], and occurs through tumor-intrinsic changes and alterations in the diverse immune and non-immune cell types that make up the tumor microenvironment (TME)[11–15]. Which of these are essential for repressing immune responses to cytotoxic chemotherapy is of significant interest, and has both preclinical and clinical relevance.

Chemotherapy has the potential to overcome some of the barriers against an effective anti-tumor immune response by stimulating the production of cytokines, chemokines, and other damage signals that recruit immune cells into the TME and prime innate and adaptive immune responses. However, immunogenic chemotherapy only disables some immune-evasive mechanisms. A promising therapeutic strategy is combining chemotherapy with blockade of immune-checkpoint proteins and immunosuppressive cytokines and metabolites, such as IL-10, IDO, and other chemokines that influence the activity of immune cells present in the microenvironment[16]. Simultaneously targeting multiple TME evasive mechanisms may potentially improve the treatment of cancer.

IL-6 is a pleiotropic cytokine frequently found in diverse TMEs. IL-6 is involved in the regulation of tissue repair, the acute-phase response[17], and is indispensable for the initiation of both innate and adaptive immune responses in many contexts[18,19]. IL-6 has been implicated in tumor development and resistance to therapy in diverse cancer types, including through its effects on the immune system[20–27]. In addition to its well-established pro-inflammatory effects, IL-6 has also been suggested to have pro-resolving and anti-inflammatory properties, and chronic IL-6 activity can stimulate immune-suppressive signals

and impair the generation of a robust immune response[18,19,28,29]. Consequently, the ultimate impact of IL-6 on the generation of anti-tumor immune responses in vivo remains unclear.

Our previous studies have identified that TME-derived IL-6 is acutely induced following chemotherapy treatment, activating cancer cell anti-apoptotic signaling and shielding lymphoma cells from cell death[20,21]. Here, we find that production of IL-6 by the TME regulates chemotherapy efficacy in ALL by inhibiting anti-leukemia immunity. Using a syngeneic mouse model of ALL, we show that the wild-type tumor microenvironment is immuno-suppressive. The immunogenic-chemotherapeutic doxorubicin extends survival through the direct induction of tumor cell death in wild-type mice, similar to its impact in patients with B-ALL[7,8], but fails to promote durable anti-cancer immunity. Interestingly, we find that IL-6 knock-out (KO) mice treated with doxorubicin completely clear leukemic cells, with the majority of these mice undergoing T-cell-dependent anti-leukemia immune responses and developing lasting immunologic memory. Thus, the presence or absence of IL-6 dictates doxorubicin efficacy by shifting its mechanism of anti-cancer clearance, which becomes primarily immunogenic in the absence of IL-6. Our results suggest that the inhibition of IL-6 may be a broadly effective therapeutic strategy to promote durable responses to standard of care genotoxic drug regimens.

## Results

**Wild-type B-ALL-bearing mice are resistant to doxorubicin treatment**. To explore the mediators of immune suppression and resistance to immunogenic chemotherapy, we used a transplantable syngeneic mouse model of BCR-ABL[+] B-ALL[30]. Transplanted leukemia cells are found primarily in the bone marrow (BM), blood, and spleen, recapitulating the relevant tissue microenvironments in the human disease[31,32]. To investigate the response of this leukemia to immunogenic chemotherapy, wild-type (WT) leukemia-bearing recipients were treated with doxorubicin (DOX) and monitored for survival. Overall survival was extended in tumor-bearing mice treated with doxorubicin, but all mice ultimately relapsed with chemoresistant disease (Fig. 1a)—a phenotype that parallels treatment failure in the clinical setting[31,32]. To further investigate the immunogenicity of this model in its native tumor microenvironment, we transplanted B-ALL tumor cells into Rag-2 KO recipient mice, which lack functional T and B cells. We found that lack of T and B cells did not significantly affect survival (Fig. 1b). CD8[+] T-cell depletion in WT mice failed to yield a statistically significant impairment in doxorubicin response but did show a small numerical difference in survival (Supplementary Fig. 1a, b). Taken together, these data suggest that the improved survival following treatment is primarily a direct cytotoxic effect of chemotherapy and is largely independent of sustained anti-leukemia immune responses.

**IL-6 promotes resistance to cytotoxic therapy**. We have previously shown that TME-derived IL-6 modulates resistance to genotoxic chemotherapy in a mouse model of Burkitt's lymphoma[20,21]. To understand the effect that loss of IL-6 in the tumor microenvironment has on leukemia response to chemotherapy in B-ALL, we transplanted leukemia cells into syngeneic IL-6 KO mice[17] and treated these mice with doxorubicin. Surprisingly, we found that doxorubicin-treated mice lacking IL-6 in the tumor microenvironment live significantly longer than WT-treated mice, with a majority of mice appearing to be cured of their disease (Fig. 1c). In the absence of treatment and shortly after treatment, we see no difference in leukemia tumor burden between WT or IL-6 KO mice. However, leukemic cell burden is significantly reduced in doxorubicin-treated IL-6 KO mice by

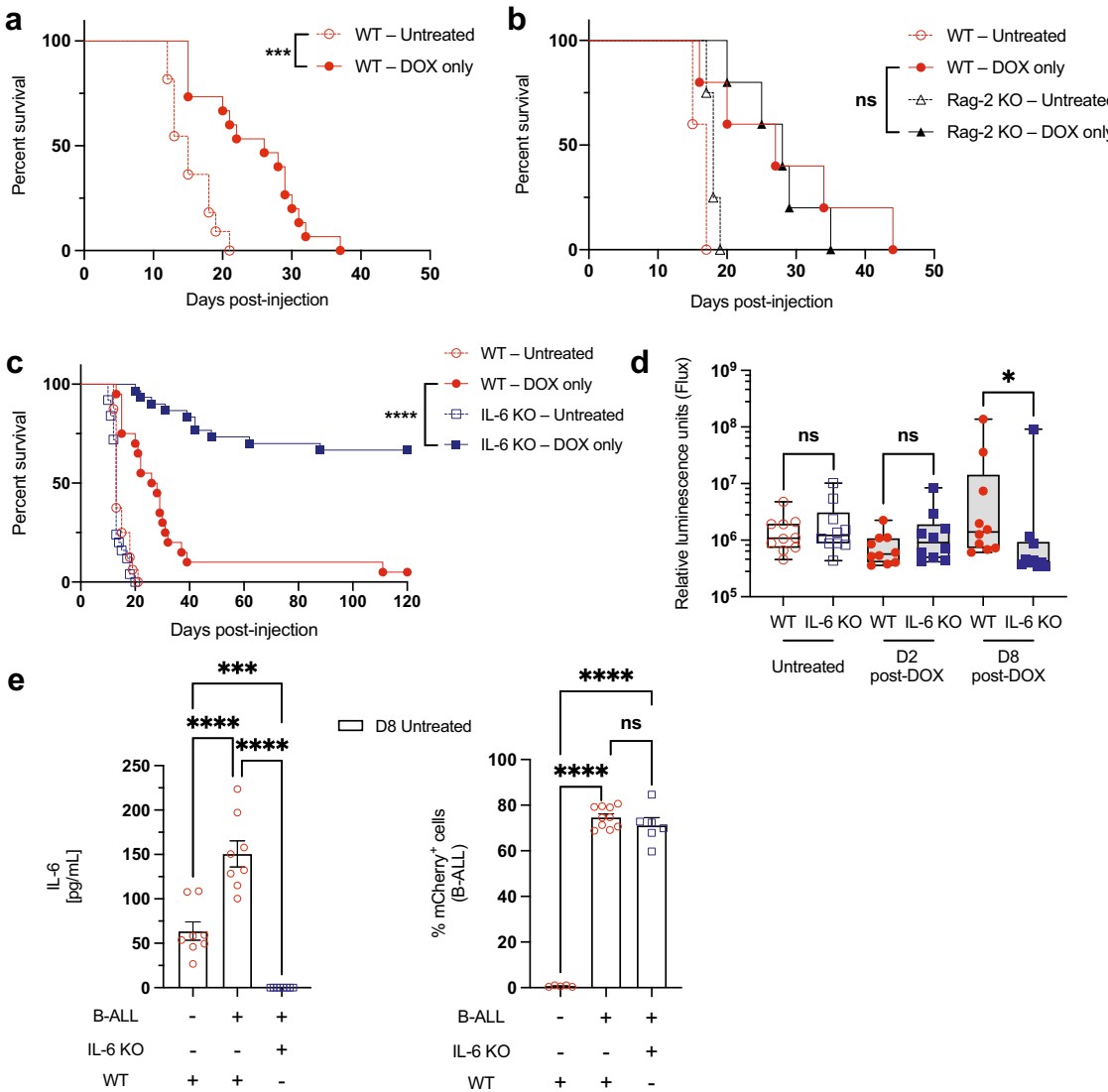

**Fig. 1 Doxorubicin extends survival of WT leukemic mice but elicits long-term disease elimination in the absence of IL-6. a** A Kaplan–Meier survival curve showing leukemic WT mice, either treated with doxorubicin (DOX) or untreated. $n = 11$ for WT-untreated, $n = 15$ for WT-treated. ***$p = 0.0001$. **b** A Kaplan–Meier survival curve showing leukemic WT or immunodeficient Rag-2 KO mice, either treated with doxorubicin or untreated. $n = 5$ per cohort, except $n = 4$ for Rag-2 KO-untreated. **c** A Kaplan–Meier survival curve showing leukemic WT or IL-6 KO mice, either treated with doxorubicin or untreated. $n = 16$ for WT-untreated, $n = 20$ for WT-treated, $n = 25$ for IL-6 KO-untreated, $n = 30$ for IL-6 KO-treated. Data from 4 independent experiments are shown. ****$p < 0.0001$. **d** A graph showing leukemia burden in vivo monitored by bioluminescence imaging. Leukemia burden in WT and IL-6 KO mice at various times before and after treatment. $n = 10$ per cohort. *$p = 0.0115$ by two-tailed Mann–Whitney test. **e** Left: a graph showing the concentration of IL-6 present in the bone marrow of tumor-free or B-ALL-bearing mice. $n = 8$ per cohort. Right: a graph showing mCherry$^+$ B-ALL percentages in the bone marrow of tumor-free or B-ALL bearing mice, quantified by flow cytometry. $n = 10$ for WT-ALL-untreated, $n = 5$ for WT-PBS-untreated, $n = 6$ for IL-6 KO-ALL-untreated. Data are represented as mean ± SEM. Shown are individual biological replicates. ***$p = 0.0008$, ****$p < 0.0001$ by Ordinary one-way ANOVA test. Log-rank (Mantel-Cox) tests were used to compare Kaplan–Meier survival curves. Boxplots show the median as the center lines, upper and lower quartiles as box limits, and whiskers represent maximum and minimum values. D2 = Day 2, D8 = Day 8. Source data are provided as a 'Source data' file.

8 days after treatment (Fig. 1d, and Supplementary Fig. 2a). Chemotherapy induction regimens against human B-ALL include corticosteroids, which have anti-inflammatory properties that could interfere with anti-tumor immune responses. Therefore, we administered doxorubicin and dexamethasone to tumor-bearing IL-6 deficient mice but failed to see any significant negative effect on anti-tumor immunity (Supplementary Fig. 1c). To determine how IL-6 is regulated by the presence of leukemic cells and whether leukemic cells or other microenvironmental cells are the primary source of IL-6 production, we transplanted B-ALL cells into WT and IL-6 KO hosts and harvested bone marrow samples

to quantify the levels of IL-6 by ELISA. While B-ALL cells do not produce IL-6 themselves, they cause an upregulation in IL-6 production by the tumor microenvironment, regardless of comparable tumor burdens in both WT & IL-6 KO mice (Fig. 1e).

Anthracyclines like doxorubicin are reported to induce cancer cell death programs that are immunogenic and prompt anti-tumorigenic host responses. To further characterize the dependence of our phenotype on ICD, we explored how doxorubicin regulates immunogenic DAMPs associated with ICD[2,5,33,34]. Analysis of CRT signal on the surface of treated leukemia cells shows that doxorubicin, but not imatinib, a BCR-ABL inhibitor

which is not known to induce ICD, induces CRT surface exposure (Supplementary Fig. 2b, d). Unexpectedly, other classic hallmarks of ICD are not significantly induced after doxorubicin treatment in this model (Supplementary Fig. 2c).

**IL-6 does not signal directly to leukemia cells to affect survival.** We next sought to understand whether IL-6 could directly promote therapeutic resistance in B-ALL cells. IL-6 signals through a receptor complex composed of the membrane-embedded signal transducer gp130 and either transmembrane or soluble forms of the IL-6 receptor[24,28]. Cells do not have to express the IL-6 receptor (IL-6R) to engage in IL-6-mediated signaling but can activate signaling from binding of soluble IL-6R (sIL-6R)-IL-6 complexes to gp130. The IL-6R was not detected on leukemia cells either in vitro (Fig. 2a) or in vivo (Fig. 2b), but is expressed on many stromal cells in the BM microenvironment (Fig. 2b). To test whether IL-6 can directly mediate resistance to doxorubicin, we cultured leukemia cells in the presence of IL-6, sIL-6R, or both IL-6 and sIL-6R to simulate signaling through sIL-6R-IL-6 complexes. Surprisingly, none of these conditions altered the sensitivity of leukemic cells to doxorubicin (Fig. 2c), suggesting that IL-6 does not directly promote resistance to doxorubicin in this system.

We have previously shown that IL-6 regulates the production of a number of other cytokines and growth factors in the bone marrow[27], leading to elevated levels of IL-10, IL-12, IL-15, and GM-CSF. To determine if these cytokines and growth factors can directly mediate resistance to doxorubicin, we cultured leukemia cells in the presence of these other cytokines. Interestingly, growth of leukemia cells in the presence of these cytokines or growth factors also had no impact on the cells' sensitivity to doxorubicin in vitro (Fig. 2d). Co-culture of leukemia cells with bone marrow stromal cells from WT or IL-6 KO mice (Fig. 2e) also did not have an effect on the cells' sensitivity to doxorubicin. These results suggest that the resistance conferred by IL-6 does not result from direct regulation of any soluble factor downstream of IL-6 signaling. We also do not observe significant levels of phosphorylated STAT3 (p-STAT3), a major signaling pathway downstream of the IL-6R, in leukemia cells in vivo before treatment (Fig. 2f) or significant differences in gene expression downstream of STAT3 in leukemic cells grown in WT and IL-6 KO mice (see below). Interestingly, while stromal p-STAT3 levels increase in response to doxorubicin treatment, there are no differences in p-STAT3 levels between WT and IL-6 KO mice at the times tested (Fig. 2f), although some change in pathway gene expression is noted (see below). To further understand the molecular mechanisms that mediate treatment resistance by IL-6, we performed immunoblot analysis of various IL-6 effectors from B-ALL-bearing bone marrow lysates. Activation of S6 kinase (S6K), a target of PI3K/mTOR signaling, was not significantly changed by the absence of IL-6 nor exposure to doxorubicin treatment. Similarly, activation of ERK1/2, a target of Ras/MAPK signaling, remained unchanged regardless of treatment conditions (Supplementary Fig. 3a-d). In addition, we were not able to detect release of the sIL-6R in co-culture of leukemia cells with bone-marrow stromal cells from WT or IL-6 KO mice (Supplementary Fig. 3e). Thus, the therapeutic benefit we see in vivo appears to be independent of IL-6 activity directly on the cancer cells and most likely mediated by its impact on the stroma.

**Doxorubicin induces immune cell infiltration into the leukemic bone marrow.** Immunogenic cell death-released immune-activating factors serve to recruit immune cells to sites of damage and activate downstream inflammatory signaling that can further recruit additional immune-cell subsets to the inflamed tissue,

spurring anti-cancer immunity[2]. To understand the role that doxorubicin has on immune cell recruitment to major sites of leukemia burden like the bone marrow and spleen, we profiled immune cell composition in leukemia-bearing mice before and after doxorubicin treatment. Before treatment, T-cells make up a small portion of cells in the bone marrow (Fig. 3a, Supplementary Fig. 3f, and Supplementary Table 1) but are much more prevalent in the spleen (Supplementary Table 2). This suggests that the bone marrow, which is the primary site of residual leukemia after treatment, may be a T-cell exclusionary microenvironment[35]. Interestingly, doxorubicin treatment selectively promotes T-cell influx into the bone marrow, but not the spleen, with increased cytotoxic and helper T-cell subsets observed in both WT and IL-6 KO mice (Fig. 3b, c, and Supplementary Table 1). We find relatively low levels of $CD3^+$–$CD4^+$–$CD25^+$ cells in the leukemic bone marrow (Fig. 3d), a subset that includes T-regulatory (T-Reg) cells. These cell populations were subtly changed after doxorubicin treatment, suggesting that the T-cell recruitment promoted by doxorubicin is cell-type specific and that doxorubicin may increase the CTL/T-reg ratio in the bone marrow, a ratio that is positively associated with survival in multiple cancer types[36,37].

In addition, doxorubicin promotes increased $CD11c^+$–$MHC$-$II^+$ dendritic cell (Fig. 3e) and $F480^-$–$CD11b^+$–$Gr$-$1^+$ neutrophil infiltration in the bone marrow (Fig. 3f). There are no major changes in the overall percentages of $CD11b^+$–$Gr$-$1^+$ cells (Fig. 3g). This latter population includes multiple mature and immature myeloid cell subsets which make up a major portion of the cells in the bone marrow. At these early timepoints after doxorubicin treatment, there is no significant difference in leukemic cell burden in the BM of IL-6 KO and WT mice (Fig. 3h), suggesting that the DNA damage induced by this agent may not account for its entire anti-tumor activity. Collectively, these data indicate that the bone marrow is an exclusionary environment for leukemia-reactive T cells. Doxorubicin treatment leads to increased dendritic and T-cell infiltration, potentially contributing to leukemia recognition and clearance in the right environmental context.

**Leukemia clearance in IL-6 KO mice is dependent on T-cell-mediated anti-tumor immune responses.** The inability of IL-6 to directly promote doxorubicin resistance stands in contrast with the increased efficacy of doxorubicin chemotherapy in IL-6 KO mice. This increased efficacy, and the T-cell influx we see after doxorubicin treatment, led us to investigate whether IL-6 might affect therapeutic response through modulation of the immune system. To study the role of T cells in the durable responses observed in IL-6 KO mice, we depleted T cells through the injection of anti-CD4 and CD8 antibodies. While T-cell-depleted IL-6 KO mice exhibit similar initial responses to doxorubicin 2 days after treatment, these mice fail to fully clear their leukemic burden, rapidly relapse, and do not exhibit the long-term survival typically seen after doxorubicin treatment of IL-6 KO mice (Fig. 4a, b). These results suggest that T-cell anti-tumor activities are essential for the profound responses to doxorubicin seen in IL-6 KO mice. Depletion of $CD8^+$ or $CD4^+$ cells alone recapitulated the effect seen with combined CD4- and CD8-depletion (Supplementary Fig. 1d, e), suggesting that long-term survival of doxorubicin-treated IL-6 KO mice is dependent on both $CD8^+$ cytotoxic T-lymphocyte (CTL) and $CD4^+$ helper activity. These data indicate that doxorubicin has the potential to promote an anti-tumor immune response, likely in part through the recruitment of T cells into the BM, but that this response is suppressed in WT mice through the production of IL-6. Next, to evaluate whether IL-6 KO mice develop lasting immunologic memory after doxorubicin treatment, we

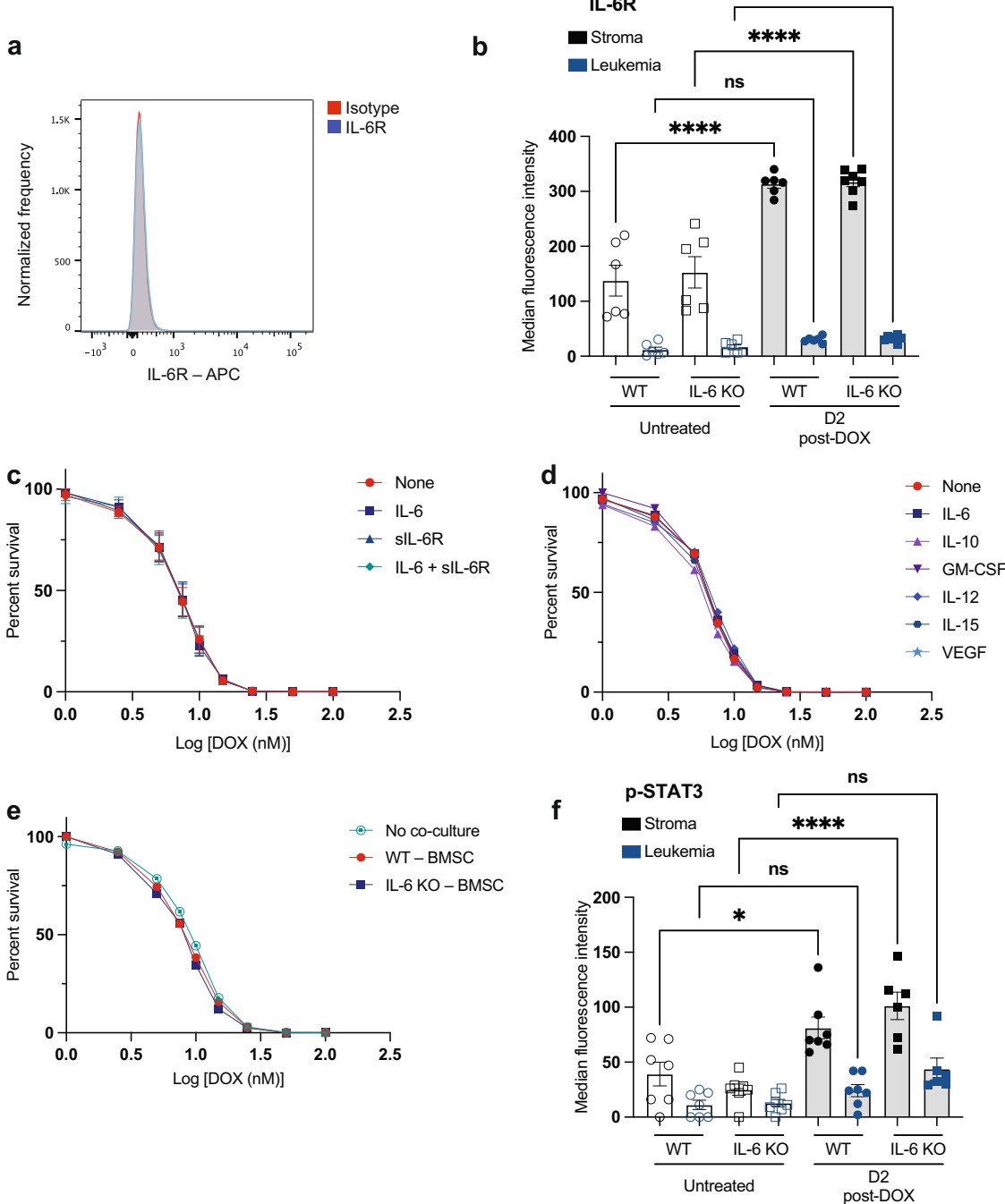

**Fig. 2 IL-6 does not promote intrinsic B-ALL chemoresistance. a** A flow cytometry plot showing IL-6R expression on the surface of leukemic cells in vitro. Histograms for IL-6R and isotype control-stained cells are overlaid. **b** A graph showing IL-6R expression in the bone marrow of leukemia-bearing mice pre- and post-doxorubicin (DOX) treatment. Data were quantified by flow cytometry. mCherry⁺ leukemia cells were used to distinguish stromal and leukemia cells. Median APC intensity from IL-6R stained cells minus isotype control-stained cells was calculated. Data from 2 independent experiments are shown, and represented as mean ± SEM. $n = 6$ per cohort, except $n = 7$ for both IL-6 KO-D2 post-doxorubicin stroma and IL-6 KO-D2 post-doxorubicin leukemia samples. ****$p < 0.0001$ by Ordinary one-way ANOVA test. **c** A dose–response curve showing leukemic cell viability in response to doxorubicin treatment in the presence or absence of IL-6 and/or sIL-6R. Viable cells were counted by flow cytometry 48 h after the addition of doxorubicin. Data from 4 independent experiments are shown, and represented as mean ± SEM. **d** A dose–response curve showing leukemic cell viability in the presence or absence of cytokines previously observed to be regulated by IL-6. Cells were treated as in (**c**) with the indicated cytokines. **e** A dose–response curve showing leukemic cell viability of cells co-cultured with bone marrow stromal cells (BMSC) from WT or IL-6 KO mice in response to doxorubicin treatment as in (**c**). **f** A graph showing p-STAT3 levels in the bone marrow of leukemia-bearing WT and IL-6 KO mice. There were no significant statistical comparisons between corresponding WT and IL-6 KO samples. Data are shown as in (**b**), from 2 independent experiments, and represented as mean ± SEM. $n = 7$ per cohort, except $n = 6$ for both IL-6 KO-D2 post-doxorubicin stroma and IL-6 KO-D2 post-doxorubicin leukemia samples. *$p = 0.011$, ****$p < 0.0001$ by Ordinary one-way ANOVA test. D2 = Day 2. Source data are provided as a 'Source data' file.

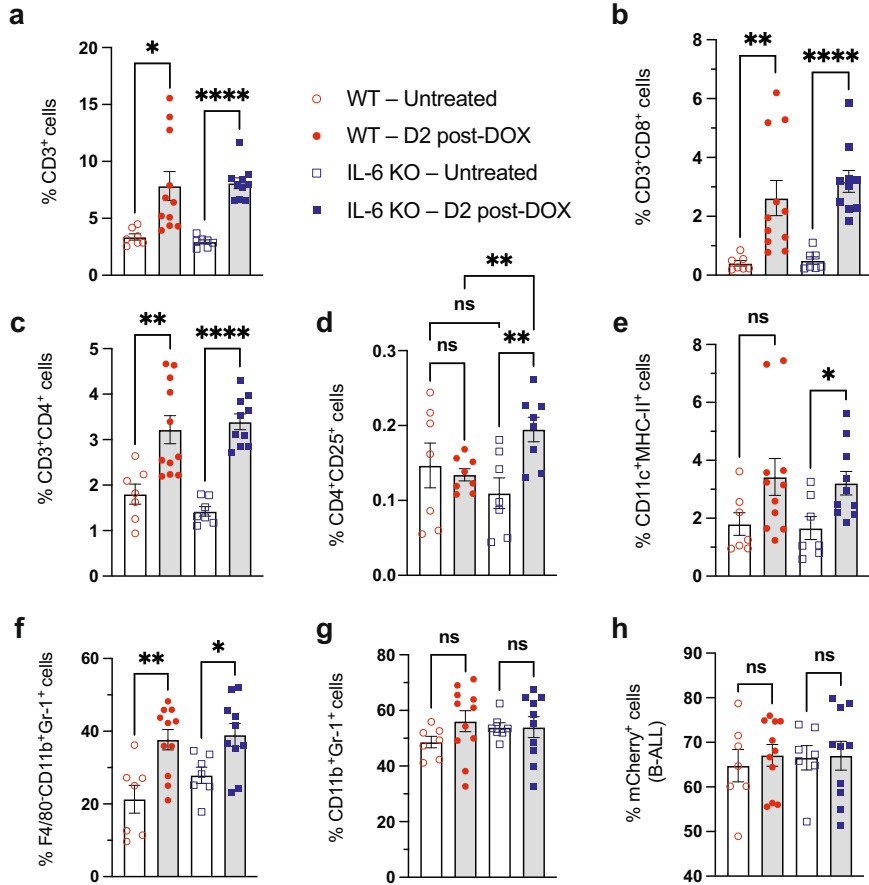

**Fig. 3 Doxorubicin induces immune cell recruitment to the TME. a** A graph showing the percentages of T cells (CD3+) in the bone marrow of leukemia-bearing mice pre- and post-doxorubicin (DOX) treatment. *$p = 0.0142$, ****$p < 0.0001$. **b** Graph showing the percentages of cytotoxic T cells (CD3+–CD8+) as in (**a**). **$p = 0.0099$, ****$p < 0.0001$. **c** Graph showing the percentages of helper T cells (CD3+–CD4+) as in (**a**). **$p = 0.0044$, ****$p < 0.0001$. **d** Graph showing the percentages of a subset of T cells (CD3+–CD4+–CD25+) as in (**a**). $n = 7$ for WT-untreated, $n = 8$ for WT-D2 post-doxorubicin, $n = 7$ for IL-6 KO-untreated, $n = 8$ for IL-6 KO-D2 post-doxorubicin mice. Data from 3 independent experiments are shown. **$p = 0.0059$ between IL-6 KO samples, **$p = 0.005$ between treated WT and IL-6 KO samples. **e** Graph showing the percentages of dendritic cells (CD11c+–MHC-II+) as in (**a**). *$p = 0.019$. **f** Graph showing the percentages of neutrophils (F480−–CD11b+–Gr-1+) as in (**a**). *$p = 0.0189$, **$p = 0.0029$. **g** Graph showing the percentages of myeloid-derived suppressor cells/monocytes (CD11b+–Gr-1+) as in (**a**). **h** A graph showing mCherry+ B-ALL percentages in the bone marrow of leukemia-bearing mice pre- and post-doxorubicin treatment. All data were quantified by flow cytometry. Data are represented as a percent of DAPI-negative (live), mCherry-negative (non-leukemic) cells for immune populations. Data for all panels are represented as mean ± SEM. $n = 7$ for WT-untreated, $n = 11$ for WT-D2 post-doxorubicin, $n = 7$ for IL-6 KO-untreated, $n = 10$ for IL-6 KO-D2 post-doxorubicin mice, and data from 4 independent experiments are shown; applies for all panels unless otherwise noted. Analyzed by two-tailed Student $t$-test. There were no significant statistical comparisons between 'untreated' and 'DOX treated' samples of different genetic backgrounds, unless shown. D2 = Day 2. Source data are provided as a 'Source data' file.

re-transplanted leukemia cells into previously cured IL-6 KO mice or naive controls (Fig. 4c) and monitored leukemia progression. Strikingly, previously cured mice were completely resistant to leukemia initiation upon tumor re-transplantation (Fig. 4d, e). These results suggest that IL-6 absence allows for the generation of lasting anti-cancer immunity that is mainly mediated by T-lymphocyte responses.

**IL-6 absence impacts diverse immune-modulatory pathways.** To further investigate the differences between WT and IL-6 KO mice, B-ALL and stromal cells were sorted from the bone marrow and RNA-sequencing was performed (Supplementary Fig. 4a). DESeq2 was used to identify differentially expressed genes in the tumor and stroma of IL-6 KO mice relative to wild-type, and rank list genes by t-statistic. GSEA analysis of the pre-ranked list using the cancer 'Hallmarks' collection from MSigDB identified few differentially regulated sets, but showed the gain of inflammatory response genesets in IL-6 KO samples, suggesting a global

difference in immune states between WT and IL-6 KO mice (Supplementary Fig. 4b). This elevated immune signature in tumor stroma seems to include increased expression of genes that are pathway components or recognized targets of IL-6 signaling, implying a potential compensatory response to decreased IL-6 pathway flux. The directionality of gene expression changes in these samples indicates that IL-6 KO leukemic mice are poised to generate an enhanced immune response, fitting our experimental data. Next, we analyzed underlying expression of enriched genesets identified by GSEA to determine if these inflammatory responses are more prominent in tumor or stroma samples. We found the most variance between IL-6 KO and WT mice pertained to the stroma samples, for both global normalized gene expression (Supplementary Fig. 4c) and for the genesets within the GSEA Hallmarks collection (Supplementary Fig. 4d-g). These results support our prior data indicating that the primary differences in IL-6 KO mice relative to WT arise from the bone marrow stroma. Rather than acting directly on tumor cells, IL-6

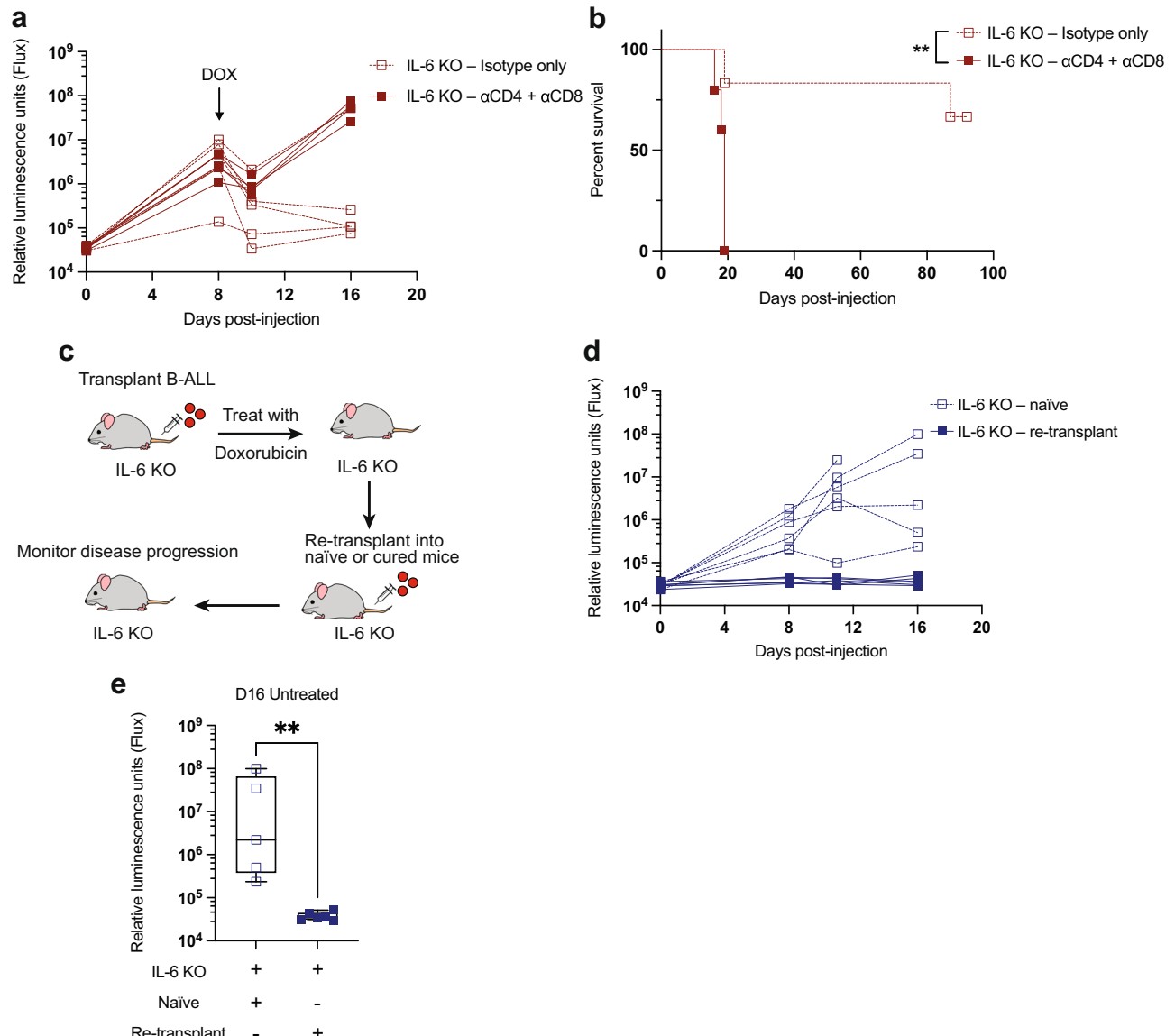

**Fig. 4 T-cell-dependent anti-tumor immunity develops after doxorubicin treatment of IL-6 KO leukemic mice. a** A graph showing leukemia burden in vivo monitored by bioluminescence imaging. CD4 and CD8 cells were depleted with antibodies and response to doxorubicin (DOX) treatment was followed. $n = 5$ per cohort. **b** A Kaplan–Meier survival curve showing leukemic IL-6 KO mice treated as in (**a**). $n = 6$ for IL-6 KO-isotype, $n = 5$ for IL-6 KO-depleted. \*\*$p = 0.0059$ by Log-rank (Mantel-Cox) test. **c** IL-6 KO mice previously cured (living > 80 days) by doxorubicin treatment were re-transplanted with leukemia cells and disease progression monitored by bioluminescence imaging in the absence of further treatment. **d** A graph showing leukemia burden in vivo in control and re-transplanted leukemia-bearing mice. $n = 6$ per cohort. **e** A graph showing leukemic IL-6 KO mice as in (**e**), D16 after disease transplant in the absence of treatment. At this time point, $n = 5$ for IL-6 KO-naive, $n = 6$ for IL-6 KO-re-transplant. \*\*$p = 0.0043$ by two-tailed Mann–Whitney test. Boxplots show the median as the center lines, upper and lower quartiles as box limits, and whiskers represent maximum and minimum values. D8 = Day 8, D16 = Day 16. Source data are provided as a 'Source data' file.

deficiency appears to alter the bone marrow stroma to broadly create a permissive immune microenvironment.

**IL-6 deficiency synergizes with anti-PD-L1 therapy to treat leukemia.** PD-1 and other immune-checkpoint proteins that play key roles in the suppression of anti-cancer immune responses are induced during T-cell activation[38]. It is thought that these proteins exist to restore normal homeostasis after an immune stimulus, preventing hyperactive immune responses and autoimmunity[39]. Cancer cells often express high levels of inhibitory checkpoint ligands and exploit the presence of these proteins on T cells to inhibit their activity. T cells in the bone marrow

of IL-6 KO mice have reduced surface expression of PD-1 (Fig. 5a). Combination treatment of WT B-ALL-bearing mice with doxorubicin and PD-L1 antibody-blocking therapy reduced leukemia burden in a subset of mice (Fig. 5b) and increased their survival (Fig. 5c). These observations suggest that higher expression of PD-1 inhibitory signals present in IL-6 proficient microenvironments might contribute to the failure of immunogenic therapy. This model of B-ALL expresses high levels of the checkpoint ligand PD-L1 (Fig. 5d), and PD-L1 expression has previously been implicated in B-ALL resistance to immune-stimulating therapy[40]. To determine whether IL-6 loss could also enhance the efficacy of PD-L1 blockade and promote anti-leukemia immune responses, we treated WT and IL-6 KO

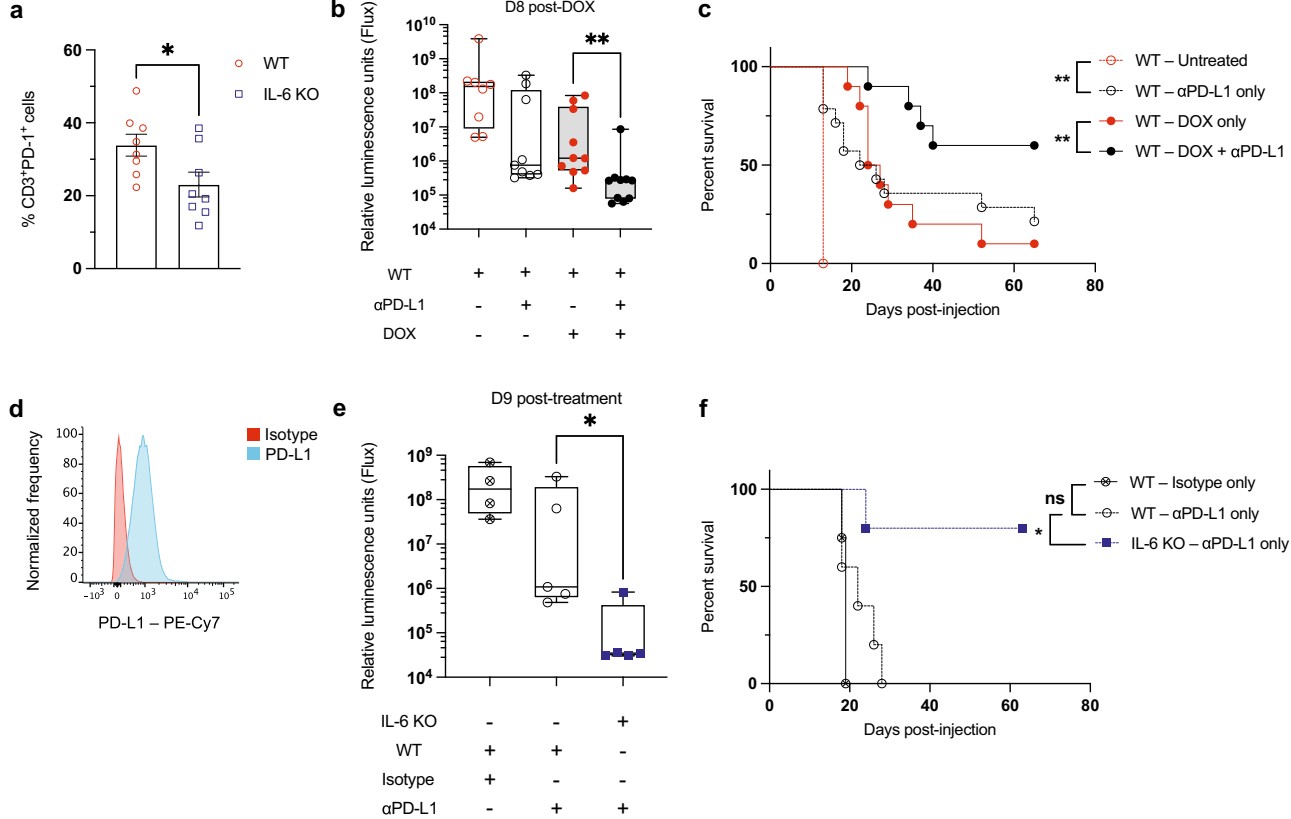

**Fig. 5 Combination treatment with doxorubicin and PD-L1 inhibition increases mouse survival, while IL-6 inhibits the efficacy of PD-L1 inhibition. a** A graph showing CD3$^+$ T-cell-PD-1$^+$ percentages in the bone marrow of tumor-free WT mice in the absence of treatment. *$p = 0.032$ by two-tailed Student $t$-test. **b** A graph showing leukemia burden in vivo monitored by bioluminescence imaging. $n = 10$ per cohort, except $n = 9$ for both WT-untreated and WT-αPD-L1. **$p = 0.0029$ by two-tailed Mann–Whitney test. **c** A Kaplan–Meier survival curve showing leukemic WT mice. $n = 10$ per cohort, except $n = 5$ for WT-untreated, $n = 14$ for WT-αPD-L1. **$p = 0.0029$ between WT-untreated and WT-αPD-L1 only, **$p = 0.0071$ between WT-doxorubicin only and WT-doxorubicin + αPD-L1, by Log-rank (Mantel-Cox) test. **d** A flow cytometry plot showing PD-L1 expression on the surface of leukemic cells in vitro. **e** A graph showing leukemia burden in vivo monitored by bioluminescence imaging. $n = 5$ per cohort, except $n = 4$ for WT-isotype. *$p = 0.0317$ by two-tailed Mann–Whitney test. **f** A Kaplan–Meier survival curve showing leukemic WT and IL-6 KO mice. $n = 5$ per cohort, except $n = 4$ for WT-isotype. *$p = 0.0126$ by Log-rank (Mantel-Cox) test. Boxplots show the median as the center lines, upper and lower quartiles as box limits, and whiskers represent maximum and minimum values. D8 = Day 8, D9 = Day 9. Source data are provided as a 'Source data' file.

leukemic mice with PD-L1 inhibitors and monitored disease progression and survival. While PD-L1 blockade exhibits modest efficacy in only a subset of WT mice, IL-6 KO mice undergo nearly complete leukemia eradication by 9 days after the initiation of PD-L1 blockade (Fig. 5e). Almost all of the PD-L1-treated IL-6 KO mice underwent durable remissions and 80% remained alive without apparent disease more than 60 days after injection (Fig. 5f). These data further suggest that production of IL-6 is a major barrier to the efficacy of immune-stimulating therapy in leukemia and that some, but not all, of its impact occurs through the regulation of T-cell PD-1 expression.

**Doxorubicin induced-immunity extends survival in tumor-bearing mice treated with IL-6 receptor blockade.** To determine the potential clinical relevance of our observations in IL-6 KO mice, we next examined the efficacy of doxorubicin treatment when combined with IL-6R blockade in WT animals bearing B-ALL. After optimizing the dosage and administration schedule of the IL-6R inhibitor (Supplementary Fig. 5a), we observed that combination treatment with doxorubicin and inhibition of IL-6 signaling with therapeutic antibodies significantly extended the survival of WT mice (Fig. 6a). Notably, 48 h after doxorubicin treatment, leukemic cell death has started to occur in both 'anti-Isotype' and 'anti-IL-6R' treatment combination groups. However, a week after doxorubicin

administration there is significantly more leukemic cell clearance in mice treated with IL-6 receptor blockade (Fig. 6b). In contrast, MC38-bearing mice were refractory to combination therapy (Supplementary Fig. 5b).

Others have shown that signaling downstream of IL-6 is important for the development, progression, and therapy response of many cancers[28], including pancreatic ductal adenocarcinoma (PDAC)[22,41,42]. Therefore, we assessed if the combination of cytotoxic chemotherapy and IL-6R blockade might also promote tumor control in a preclinical model of PDAC subcutaneously injected into WT mice. Once tumors were established, mice received combination treatment with doxorubicin and IL-6R therapeutic antibodies. Expression of IL-6R was detected in the stromal cells in vivo (Supplementary Fig. 5c). Consistent with our observations in B-ALL, combination treatment with doxorubicin and αIL-6R had significant inhibition of PDAC tumor growth (Fig. 6c, d). Similarly, doxorubicin treatment had significant inhibition of PDAC tumor growth in IL-6 KO mice (Fig. 6e, f). Intriguingly, p-STAT3 levels from bulk PDAC tumor samples do not significantly change between WT and IL-6 KO mice, nor in response to doxorubicin treatment at the times examined (Supplementary Fig. 5d).

Finally, we re-transplanted PDAC cells, 5 days after doxorubicin treatment, into the opposite flanks of PDAC-bearing IL-6

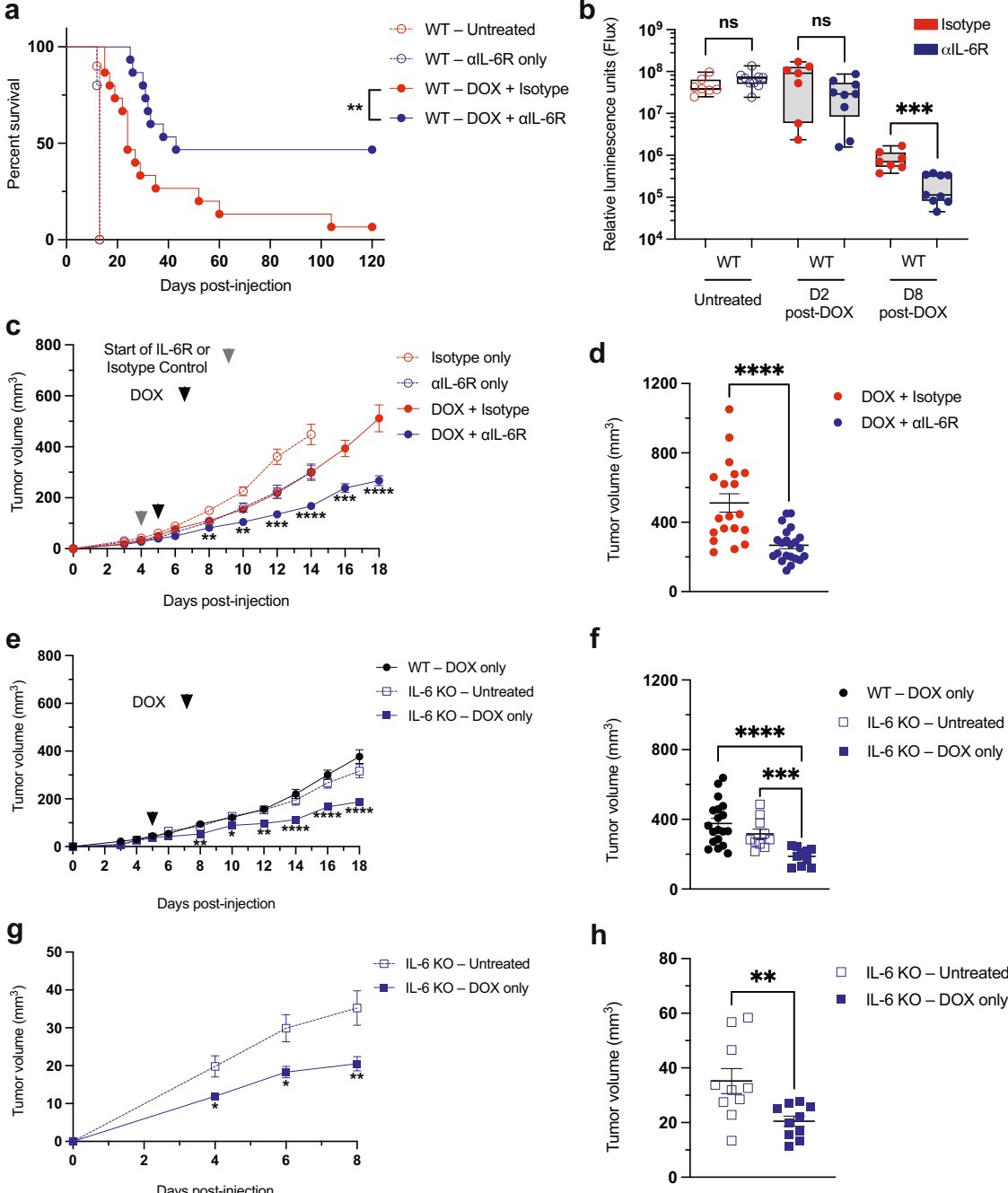

**Fig. 6 Therapeutic IL-6 inhibition enhances the efficacy of doxorubicin treatment. a** A Kaplan–Meier survival curve of B-ALL-bearing mice. $n = 15$ for both doxorubicin (DOX) + Isotype and doxorubicin + αIL-6R treated, $n = 10$ for untreated, and $n = 5$ for αIL-6R only treated mice. **$p = 0.0062$ by Log-rank (Mantel-Cox) test. **b** A graph showing leukemia burden in WT mice treated with doxorubicin and either an IL-6R blocking antibody or isotype control. Tumor burden was monitored by bioluminescence imaging. $n = 7$ for doxorubicin + Isotype, $n = 9$ for doxorubicin + αIL-6R treated mice. ***$p = 0.0002$ by two-tailed Mann–Whitney test. **c** A graph showing PDAC tumor burden in WT mice treated with doxorubicin and either an IL-6R blocking antibody or isotype control. $n = 10$ for Isotype only, $n = 15$ for αIL-6R only, $n = 19$ for doxorubicin + Isotype, and $n = 22$ for doxorubicin + αIL-6R. Data from 4 independent experiments. **$p < 0.035$, ***$p < 0.005$, ****$p < 0.0001$. **d** A graph showing PDAC tumor burden on day 18 for individual animals within the indicated treatment groups. Shown are the individual biological replicates from (**c**). ****$p < 0.0001$. **e** A graph showing PDAC tumor burden in WT and IL-6 KO mice treated with doxorubicin or untreated. $n = 10$ per cohort, except $n = 19$ for WT-doxorubicin only. Data from 1 independent experiment. *$p < 0.05$, **$p < 0.035$, ****$p < 0.0001$. **f** A graph showing PDAC tumor burden on day 18 for individual animals within the indicated treatment groups. Shown are the individual biological replicates from (**e**). ***$p = 0.0003$, ****$p < 0.0001$. **g** A graph showing PDAC tumor burden of secondary transplants (5 days after doxorubicin treatment) into IL-6 KO mice that had already received a primary PDAC transplant and were treated with doxorubicin or untreated. $n = 10$ per cohort. *$p < 0.05$, **$p < 0.035$. **h** A graph showing PDAC tumor burden on day 8 for individual animals within the indicated treatment groups. Shown are the individual biological replicates from (**g**). **$p = 0.0039$. Two-tailed Mann–Whitney tests were used to compare groups in panels (**c–h**), and data are represented as mean ± SEM. Boxplots show the median as the center lines, upper and lower quartiles as box limits, and whiskers represent maximum and minimum values. D2 = Day 2, D8 = Day 8. Source data are provided as a 'Source data' file.

KO mice (Supplementary Fig. 5e). The PDAC tumors transplanted into previously doxorubicin-treated IL-6 KO animals showed reduced growth compared to the tumors transplanted into untreated IL-6 KO hosts (Fig. 6g, h). These results suggest a role for IL-6 in preventing an active and long-lasting anti-tumor immune response. Thus, antibody-mediated inhibition of IL-6 signaling can promote durable responses to genotoxic chemotherapy in both hematopoietic and solid malignancies.

## Discussion

Genotoxic chemotherapy primarily exerts its effects via DNA damage-induced cell death[1]. However, work from multiple labs has demonstrated that a subset of commonly used chemotherapeutics can also stimulate immunity in specific contexts[2,5]. Despite this, it remains unclear the extent to which immune responses contribute to the efficacy of cytotoxic chemotherapy and the contexts in which they do so. In both mice and humans, immunogenic chemotherapy rarely promotes lasting anti-tumor immune responses. While there are many examples of the TME regulating therapeutic efficacy in vivo[15], how the TME tunes the immune responses to immunogenic cell death (ICD) is poorly understood and of broad clinical relevance. Here, we show that IL-6 controls a mechanistic switch between primarily cytotoxic cell death and immune-mediated clearance of tumor cells after genotoxic chemotherapy treatment.

We used a mouse model of acute lymphoblastic leukemia that closely recapitulates the microenvironment and therapy responsiveness of the human disease[30] to investigate the mechanisms of immune suppression after treatment with immunogenic chemotherapy. We show that while doxorubicin modestly extends animal survival in WT mice, it does not generate robust anti-cancer immunity and mice ultimately fail to clear their leukemia burden. In contrast, in the absence of IL-6, the majority of leukemic mice are cured after doxorubicin treatment in an immune-mediated fashion. This demonstrates that IL-6 is an important TME-derived paracrine factor that suppresses the generation of robust anti-tumor immunity. Consequently, we find that microenvironmental context not only impacts therapy responsiveness but alters the mechanism by which a commonly used clinical agent exerts its activity. This data indicate that the efficacy of conventional DNA-damaging therapies and their ability to induce anti-cancer immunity in human cancers may be limited by immunosuppressive factors in the TME, such as IL-6. These results highlight the role of the TME in the cancer cell's response to therapy and indicate how further study of the microenvironmental regulators of ICD could impact the clinical utility of cytotoxic chemotherapeutics.

We have previously shown that the bone marrow is a site of resistance to antibody-based therapy in double-hit lymphoma, where the immune-suppressive microenvironment impairs innate immune-mediated clearance of antibody-bound cells[43]. Interestingly, our findings here demonstrate that doxorubicin promotes T- and dendritic cell influx into the bone marrow, transforming it into a pro-immunogenic microenvironment. Major determinants of immunogenicity and ICD include the release of HMGB1 and surface exposure of CRT from dying cancer cells[44,45]. Doxorubicin-treated leukemia cells induce CRT surface exposure and HMGB1 release in the TME is preserved, although other mediators of immunogenicity in this system remain to be defined. While the pro-immunogenic conditions present after doxorubicin treatment are favorable for the clearance of leukemic cells, microenvironmental IL-6 production—which is increased by the presence of leukemic cells—suppresses the expected anti-cancer immune responses. Thus, cancer-cell induced IL-6 release in designated microenvironments may help to disguise immunogenic cell death states.

Paracrine signals produced in the TME play a major role in defining the immune context of tumors and show great potential for therapeutic manipulation. IL-6 is a pleiotropic cytokine involved in the regulation of many processes including immune activation[28], but chronic IL-6 activity can also weaken the generation of an effective immune response[19]. For example, chronic STAT3 activity downstream of IL-6 can impair the generation of new adaptive immune responses[46]. In the context of ICD, IL-6 may impair anti-cancer immunity through the creation of a microenvironment in which an acute inflammatory stimulus from cell death is less likely to generate a productive immune response. Our data suggest multiple downstream effectors are likely active in mediating the profound regulation of anti-cancer immunity we see after cytotoxic therapy. While we detected no differences in p-STAT3 protein levels between IL-6 KO and WT leukemic and PDAC-bearing mice at the times tested, our RNA-sequencing studies do show alterations in JAK/STAT pathway components in stroma from IL-6 KO mice. We also find higher T-cell PD-1 expression in the presence of IL-6 indicating a potentially more exhausted T-cell population. Future detailed interrogation of the effectors downstream of IL-6 will be necessary to elucidate the mechanisms underlying this significant clinical response.

While showing promising efficacy in a number of cancer types, immunotherapy can increase the activity of the immune system, causing a variety of inflammatory and auto-immune phenomena that instigate significant morbidity. These immune-related adverse events are commonly treated with steroids. However, there is concern that high-dose steroids may not fully help to alleviate the immune-related adverse events and, additionally, that they may blunt the anti-cancer effects of immunotherapies[47]. This emphasizes the need to find alternatives for treating immune-related adverse consequences. The humanized monoclonal anti-IL-6R antibody, Tocilizumab, has been used to treat inflammatory toxicity associated with immune-checkpoint blockade and the cytokine release syndrome (CRS) associated with chimeric antigen receptor T-cell therapy[48]. Interestingly, our data suggest that IL-6 blockade may be able to decouple auto-immune and anti-cancer immune responses, potentially increasing anti-cancer immunity while treating auto-immune toxicity. This phenomenon has recently been reported for TNF-blockade as well[49].

Given the many mechanisms by which cancer can evade immune surveillance[4,10], combination therapies that block multiple immune-suppressive mechanisms will be essential to promote responses in the majority of tumors. Consistent with this idea, leukemic WT mice treated with doxorubicin and PD-L1 inhibitors are more readily able to clear their disease when compared to single-agent treated mice. In addition, therapeutic IL-6R inhibitors can synergize with doxorubicin to eliminate leukemic cells from WT mice. Likewise, we show that this combination improves the response of PDAC tumors, suggesting that this phenomenon may extend to certain solid tumors. While IL-6 likely executes its immune-suppressive properties through the regulation of multiple immune processes, our data demonstrate that loss of IL-6 enhances the generation of anti-cancer immunity in response to multiple immune-stimulating therapies. IL-6 inhibition could help sustain the limited anti-cancer immune responses normally induced by cytotoxic agents in the clinic.

Here, we establish that three interventions, IL-6 inhibition, doxorubicin treatment, and PD-L1 blockade, each of which alone fails to promote lasting anti-leukemia immunity, achieve much more durable responses in combination. Importantly, we show that the state of the TME profoundly impacts both the efficacy and the primary mechanism of action of a commonly used cytotoxic agent. Taken together, these data suggest that

combination therapy with immunogenic chemotherapy, manipulation of the tumor microenvironment through IL-6 inhibition, and checkpoint blockade is a promising therapeutic approach for treating human cancer.

## Methods

**Cell culture and chemicals**. B-ALL cells were grown at 37 °C, 5% $CO_2$, in 500 mL RPMI, 50 mL FBS, 10 mL glutamine, 5.5 mL β-ME (5 mM), and 5 mL Pen. Strep. Luciferase[+] BCR-ABL[+] B-ALL male cells were a gift from Richard Williams[30]. To make mCherry[+] B-ALL cells, the MSCV-mCherry retroviral vector was transfected into Phoenix cells to produce retrovirus and B-ALL cells were infected in the presence of polybrene and sorted twice on a FACS-AriaIII (Becton Dickinson) to get a pure mCherry[+] population. PDAC and MC38 cells were grown at 37 °C, 5% $CO_2$, in 500 mL DMEM, 50 mL FBS, and 5 mL Pen. Strep. PDAC cells were a gift from Matthew Vander Heiden. MC38 or Colon 38 cells were acquired from the Developmental Therapeutics Program Tumor Repository at Frederick National Laboratory. All cell lines used regularly tested negative for *Mycoplasma* detection (MycoAlert Plus kit, Lonza).

**Mice and transplantation**. C57BL/6J (wild type) and C57BL/6J $Il$-$6^{-/-}$ mice, 6–8-week old, were purchased from Jackson Laboratory (RRID: IMSR_JAX:000664, and IMSR_JAX:002650). 500,000 BCR-ABL[+] B-ALL cells (mCherry[+] or negative depending on the experiment) were injected via tail vein into C57BL6/J mice of the appropriate genotype. On day 8 post-injection, mice were treated via intraperitoneal injection with 10 mg/kg doxorubicin (LC Labs) dissolved in normal saline solution. Mice were sacrificed when moribund. When applicable, mice were treated for 7 days with 50 mg/kg imatinib by oral gavage and sacrificed when moribund. For re-transplantation experiments, IL-6 KO mice previously cured of B-ALL by doxorubicin treatment were re-injected with 500,000 B-ALL cells (>100 days after initial injection) and disease burden and survival were monitored. 500,000 MC38 or PDAC cells were injected via subcutaneous injection into the hind-flanks of C57BL6/J mice. 200,000 PDAC cells were used for re-transplantations into IL-6 KO mice previously treated with doxorubicin. Subcutaneous tumor burden was measured with electronic calipers using the following formula: $1/2 \times D \times d^2$; where 'D' is the major measurable axis and 'd' is the minor axis. Maximal tumor burden/size allowed was no larger than 1 cm in any direction and no deep ulceration. On a case-by-case basis, veterinary technicians allowed exceptions of tumor sizes larger than 1 cm if no deep ulceration was present and if mice seemed alert and responsive. Mice were bred in the SPF-animal facility in the Koch Institute and the Massachusetts Institute of Technology Department of Comparative Medicine approved all procedures and animal handling for the work presented here. Animals were monitored carefully for fitness and sacrificed when moribund in accordance with institutional Committee on Animal Care (CAC) procedures. Both female and male sexes were used. Food (ProLab RMH 3000) and water were given ad libitum. Animals were housed at 68–72 ˚F, with a relative humidity of 30–70%, and a dark/light cycle of 12/12 h.

**Bioluminescence imaging**. Leukemic mice were imaged 1 day before doxorubicin treatment, the day of treatment, 2 days post-treatment, and 8- or 9-days post-treatment depending on the experiment. 165 mg/kg luciferin was injected prior to imaging and mice were anesthetized using isoflurane prior to imaging on the IVIS Spectrum-bioluminescence and fluorescence imaging system (Perkin Elmer), and analyzed with the Living Image software.

**Immune profiling**. Leukemic mice were sacrificed 8 days post-injection (untreated), 2 days after doxorubicin, or 7 days post-treatment for analysis of immune-cell infiltration in bone marrow and spleen. Bone-marrow cells from WT and IL-6 KO mice were extracted by crushing both femurs and tibias with mortar and pestle in RBC Lysing Buffer (Sigma-Aldrich, R7757) for 5 min and resuspended in 3% FBS-PBS (FACS Stain buffer). Splenic cells were extracted by crushing the spleen between glass slides into RBC Lysing Buffer and following the same protocol as above. Cells were stained with combinations of the following conjugated antibodies: CD3–FITC (17A2, BioLegend #100204; 1:100), CD4–APC (RM4-5, BD Biosciences #561091; 1:100), CD4–APC-Cy7 (GK1.5, BioLegend #100414; 1:100), CD8–PE-Cy7 (53-6.7, BD Biosciences #552877; 1:100), CD25–APC-Cy7 (PC61, BioLegend #102026; 1:100), CD69–PerCP-Cy5.5 (H1.2F3, BioLegend #104522; 1:100), CD11c–FITC (HL3, BD Biosciences #553801; 1:100), CD103–PerCP-Cy5.5 (2E7, BioLegend #121416; 1:100), CD86–APC (GL-1, BioLegend #105012; 1:100), MHC-II–APC-Cy7 (M5/114.15.2, BioLegend #107628; 1:100), MHC-II–PerCP-Cy5.5 (M5/114.15.2, BioLegend #107626; 1:100), CD11b–PE-Cy7 (M1/70, BioLegend #101216; 1:100), F4/80–APC (BM8, BioLegend #123116; 1:100), Gr-1–FITC (RB6-8C5, eBioscience #50-991-9; 1:100), IL-6R–APC (D7715A7, BioLegend #115812; 1:100), PD-1–BV421 (29F.1A12, BioLegend #135217; 1:100), MHC-I–FITC (34-1-2S, Abcam #ab95572; 1:100), MHC-II–FITC (M5/114, Abcam #ab239229; 1:100), and PD-L1–PE-Cy7 (10F.9G2, BioLegend #124314; 1:100) for 1 h at 4 °C. 3 μM DAPI was added to the last wash to determine live cells and samples were analyzed on LSR-II HTS flow cytometer

(Becton Dickinson). For all flow cytometry experiments, FlowJo was used for analysis.

**Cytokine dose response**. B-ALL cells were plated at 10,000/well in a 96-well plate. Cells were treated with ≥10 ng/mL IL-10, GM-CSF, IL-12, IL-15, VEGF, IL-6, sIL-6R, or IL-6+sIL-6R (PeproTech) and doxorubicin (LC Labs) at 100, 50, 25, 15, 10, 7.5, 5, 2.5, 1, 0.5, and 0 nM concentrations. Cell count was obtained via flow cytometry FACS Calibur HTS (Becton Dickinson) with propidium iodide used to exclude dead cells.

**Bone marrow co-culture**. Bone-marrow cells from WT and IL-6 KO mice were extracted as described above, without the use of RBC lysis buffer. Extracted cells were plated in leukemia cell medium. Washes were performed until adherent cells became confluent at which point, they were transferred to 96-well plates, adhered for 24 h, and used for co-culture dose–response experiments as described above.

*PDAC tumor dissociation*. PDAC tumors harvested from euthanized mice were placed in 2.5 mL's of 1% FBS-RPMI and manually minced with blades. 5 mL's of digestion buffer was then added and samples were incubated for 30 min in a 37 °C water bath inside gentleMACS C tubes (Miltenyi Biotec, 130-096-334). Digestion buffer was prepared as follows: 1% FBS, 0.8 M HEPES pH~7.5 (Invitrogen, 15630080), 1 mg/mL collagenase (Millipore Sigma, C2674), 4 U/mL DNAaseI (New England Biolabs, M0303), in HBSS (Millipore Sigma, 55037C). MACS tubes were then agitated with a MACS dissociator (Miltenyi Biotec, 130-093-235) for 1 min, and samples were quenched with 5 mL's FBS. Samples were then filtered through 70 and 30 μm filters (Miltenyi Biotec, 130-110-916 and 130-110-915, respectively), and spun at 1200 rpm for 10 min. Samples were washed once with PBS by repeating spinning cycle, and finally resuspended as single-cell suspensions in PBS.

**p-STAT3 stain**. Bone marrow and splenic cells from WT and IL-6 KO mice were extracted as described above, fixed in 3–4% paraformaldehyde, stained with primary p-STAT3 (Tyr705, D3A7, Cell Signaling Technology #4323S; 1:25) or IgG-isotype control (DA1E, Cell Signaling Technology #2975S; 1:25) at the same concentration. Alternatively, cells were fixed and permeabilized with a nuclear staining buffer set (Thermo Scientific, 00-5523-00), following manufacturer's instructions. Prior to fixing and permeabilization, staining of cell surfaces markers was performed with CD3–BV605 (17A2, BioLegend #100237; 1:100), and Zombie Aqua Fixable Viability Dye (BioLegend, 423102; 1:100). PhosSTOP 1X (Sigma-Aldrich, 4906837001) was used in every buffer. Cells were analyzed by flow cytometry using an LSR-II (Becton Dickinson) or LSR-Fortessa, and p-STAT3 levels were measured. Median FITC channel of isotype controls was subtracted from p-STAT3-stained samples to get p-STAT3 levels in a given cell population.

**Western blot assays**. Bone marrow cells from WT and IL-6 KO mice, untreated or doxorubicin treated, were harvested by centrifugation of dissected femur and tibia. Red blood cells were depleted from the bone marrow by a 5-min incubation in red blood cell lysis buffer (Sigma-Aldrich, R7757). Red cell lysis was quenched with PBS. PhosSTOP 1X (Sigma-Aldrich, 4906837001) was used in every buffer. A column with CD19 magnetic beads (Miltenyi Biotec, 130-121-301) was used to enrich for B-ALL cells (CD19[+]), and the CD19[−] flowthrough was regarded as the stromal cells from the leukemic bone marrow. Tissue samples were homogenized in standard RIPA buffer, with a cocktail of protease (Thermo Scientific, 87786) and phosphatase inhibitors (Sigma-Aldrich, 4906837001). Protein concentrations were measured using BCA (Fisher Scientific, 23225). Cell extracts with the same amount of protein were mixed with 6X reducing Laemmli buffer (Boston BioProducts, BP-111R), boiled at 95 °C for 5 min, and subjected to electrophoresis using 4–20% sodium dodecyl sulfate polyacrylamide gels (Bio-Rad). Proteins were transferred to methanol-activated PVDF membranes (Millipore Sigma, IPFL00010) and blocked with TBST buffer (LI-COR Biosciences, 927-66003) for 1 h at room temperature. Blots were incubated at 4 °C overnight with primary antibodies, followed by secondary antibodies conjugated with LI-COR fluorophores. Samples were scanned with an Odyssey CLx imaging system (LI-COR Biosciences). Anti-actin (13E5, Cell Signaling Technology #4970S; 1:1000), anti-S6K (R&D Systems #AF8964; 1:200), anti-p-S6K (Thr389, Cell Signaling Technology #9205S; 1:1000), anti-vinculin (E1E9V, Cell Signaling Technology #13901S; 1:1000), anti-ERK (W15133B, BioLegend #686902; 1:1000), and anti-p-ERK (Thr202/Tyr204, Cell Signaling Technology #9101S; 1:1000), anti-rat IRDye 680RD (LI-COR #926-68076; 1:5000), anti-rabbit IRDye 800CW (LI-COR #926-32211; 1:5000), anti-goat IRDye 680RD (LI-COR #926-68074; 1:5000), anti-rabbit IRDye 800CW (LI-COR #926-32213; 1:5000). Cropped and uncropped blot images are shown in Supplementary Fig. 3a-d.

**In vivo T-cell depletion and mouse antibody treatment**. WT and IL-6 KO leukemic mice were IP-injected on days 3 and 4 post B-ALL transplantation and then every 3 days thereafter with 200 μg CD4 (GK1.5, BioXCell #BE0003-1) and 200 μg CD8 (2.43, BioXCell #BE0061) depletion antibodies dissolved in sterile PBS. IL-6R Ab (15A7, BioXCell #BE0047) was injected every other day (500 μg/mouse) starting 3 days after leukemia transplantation (unless noted otherwise). PD-L1

antibody, at 200 μg/mouse (10F.9G2, BioXCell #BE0101), was injected on Days 7, 10, and 13 after disease transplantation. Rat IgG2b (LTF-2, BioXCell #BE0090) was used as an isotype control. IL-6R Ab was injected every other day (200 μg/mouse) starting 4 days after PDAC or MC38 transplantation. Randomization of animal cohorts was performed before transplantation of disease and before the start of any treatment. When able, the experimenter was blinded to the individual mice being examined, although this was not performed in all experiments. Cohorts of 5 mice per cage were used and key findings repeated in multiple independent experiments, as detailed in figure legends.

**ELISA assays.** B-ALL and bone marrow cells harvested from the same mouse, untreated or doxorubicin treated, were seeded in 6-well tissue culture plates for 24 h. The cell culture plates were centrifuged, and the supernatants collected and stored at −80 °C until measuring the cytokine levels. HMGB1 (Fisher Scientific, NBP262782), CXCL10 (Thermo Scientific, BMS6018), IL-6 (Thermo Fisher Scientific, 88-7064-88), sIL-6R (R&D Systems, MR600).

**RNA isolation from tumor stroma.** Cells were isolated from the bone marrow of wild-type and IL6-KO mice by centrifugation of dissected femur and tibia. Red blood cells were depleted from the bone marrow by a 5-min incubation in red blood cell lysis buffer (Sigma-Aldrich, R7757). Red cell lysis was quenched with PBS. Cells were centrifuged at $500 \times g$ for 5 min and supernatant was aspirated. Cells were resuspended in FACS buffer (PBS with 5% FBS and 1 μg/mL DAPI). Cells were sorted on a FACS-AriaIII (Becton Dickinson) running BD FACS Diva software. Stromal cells were isolated from B-ALL by gating for mCherry-negative cells following isolation of live singlets. Cells were collected and centrifuged at $500 \times g$, then supernatant was aspirated. Cells were snap-frozen in liquid nitrogen. Cells were thawed on ice and RNA was isolated using the RNeasy Isolation Kit (Qiagen, 74134) according to the manufacturer's instructions.

**RNA sequencing, data processing, and analysis.** RNA libraries were prepared using the NEB Ultra II ribodepletion kit (E6310) according to the manufacturer's instructions. Libraries were sequenced using an Illumina NovaSeq 6000 to a depth of $\sim 2 \times 10^7$ single-ended reads per sample with a read length of 75 nucleotides. Reads were aligned to the mm10 genome, and transcript abundance was quantified using salmon (version 1.3.0). Salmon quant command was executed with the following flags: --validateMappings --gcBias --seqBias. Read counts were normalized and differentially expressed genes were identified using the DESeq2 R package (version 1.28.1). The DESeq command was executed with default options. Genes with >10 read counts were rank-listed by t-statistic, and GSEA analysis of the pre-ranked gene list was performed using the clusterProfiler R package (version 3.16.1). The GSEA command was executed with the following options: eps = 0.0 and TERM2GENE = msigdb cancer hallmark gene set.

**Statistics and reproducibility.** GraphPad Prism9 software (GraphPad Software, Inc.) or Microsoft Excel were used to perform statistical analysis. Respective tests are indicated in the figure legends. Error bars represent mean ± SEM, unless noted otherwise. Log-rank (Mantel-Cox) tests were used to compare Kaplan–Meier survival curves. Immune infiltration between WT and IL-6 KO samples was analyzed by a two-tailed Student's t-test. For all statistical tests, α was limited to 0.05 and $p < 0.05$ was considered statistically significant. Boxplots show the median as the center lines, upper and lower quartiles as box limits, and whiskers represent maximum and minimum values. If present, outliers were included in the reported data.

**Reporting summary.** Further information on research design is available in the Nature Research Reporting Summary linked to this article.

## Data availability
The gene expression datasets generated during the current study have been deposited in the National Center for Biotechnology Information Gene Expression Omnibus (GEO)[50], and are accessible through GEO Series with the following accession number: GSE184107. The remaining data are available within the article, supplementary information, and provided source data file. Source data are provided with this paper.

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

## Acknowledgements

We are grateful to Stefani Spranger, Tyler Jacks, Boyang Zhao, and Jesse Engreitz for their advice with experiments. We acknowledge the Vander Heiden laboratory for providing reagents. We also thank the entire Hemann laboratory for helpful discussions and reagents. This project was funded in part by the Ludwig Center for Molecular Oncology at MIT, the MIT Center for Precision Cancer Medicine, and the Koch Institute Support (core) Grant P30-CA14051 from the NCI. L.R.M.B. is supported by the MIT Center for Precision Cancer Medicine, and E.H.B. was supported by the National Institutes of Health Grant (T32GM007753). This work was also supported in part by NCI R01-CA233477, R01-CA226898, and NIH/NIAID R21AI151827 to M.T.H.

## Author contributions

E.H.B. and M.T.H. conceptualized the study. E.H.B., L.R.M.B., and M.T.H. designed the study. E.H.B., L.R.M.B., I.Z., D.R.G., and J.F. acquired the data. E.H.B., L.R.M.B., I.Z., D.R.G., J.F., and M.T.H. interpreted and analyzed the data. E.H.B., L.R.M.B., and M.T.H. drafted the manuscript and figures. All authors revised, edited, and approved the final version of the manuscript and agree to be held accountable for personal contributions.

## Competing interests

The authors declare no competing interests.
