## [Peer Review File · Nature Communications]

Reviewers' Comments:

Reviewer #1:

Remarks to the Author:

The Authors have convincingly addressed most of my comments on the initial submission, and the manuscript is now significantly improved. I have no additional comments.

Reviewer #2:

Remarks to the Author:

The authors showed that doxorubicin treatments can recruit immune cells to bone marrow, and that IL6 secreted by leukemic and stromal cells induces immune suppression in bone marrow microenvironment. Furthermore, IL6 depletion in bone marrow tumor microenvironment resulted in long-term T-cell mediated anti-cancer immune response. Finally, their study suggested IL6 depletion can synergize PD1-PDL1 immunotherapy. Although the findings of the DNA damage chemotherapy induced immune cell recruitment in TME has some novel aspect, overall the results shown in the manuscript lack innovation.

While the authors attempted to address concerns raised by the three previous reviewers and improve the scientific rigor, there are still a number of concerns.

1. The detailed mechanism of IL6 suppressed T-cell immune response in this study is not clear.

They ruled out STAT3 activation in bone marrow tumor microenvironment in the B-ALL cancer model. However, IL6/IL6R/STAT3 signaling has been shown to be very important in pancreatic stromal cells (stellate cells) and PDAC tumor microenvironment. Since the authors used PDAC mouse model as their second animal model in the revised manuscript, it is unclear whether IL6-mediated cytotoxic T-cell suppression is STAT3-independent in PDAC TME.

2. The results from RNAseq are still underdeveloped, and the data they showed in the manuscript are very confusing. For example, the gene signatures in stimulatory and inhibitory T cell function gene sets both went down in IL6-KO cells in supplemental Figure 3 (or only down-regulated genes are listed?). Since the authors claimed a global difference in immune cells in TME between WT and IL6-KO mice, the ranking of all enriched gene sets, statistical significance and the enrichment scores (including FDR) from GSEA analysis should be included in the manuscript.

3. Variable numbers (and sometimes very low numbers) of mice were used in the different groups in the same experiment, which decreases the scientific rigor and statistical power.

4. For PDAC retransplant, the tumor sizes in both groups may be too small (below 40 mm³) to interpret the results. No methodology was mentioned how the tumor size was measured.

Reviewer #3:

Remarks to the Author:

The authors have addressed numerous concerns I raised previously, which have no doubt improved the quality of the work.

However, the authors have not addressed a key comment regarding mechanism, which is based on the statement "while stromal p-STAT3 levels increase in response to DOX treatment, there is no difference between WT and IL-6 KO mice (Fig. 2f), suggesting that the effects of IL-6 may occur independently of changes in STAT3 signaling". As requested before, authors should demonstrate what other well-documented signaling pathways downstream of IL-6 are playing a role by performing immunoblots/flow cytometry and using pathway-specific inhibitors.

Also, even though they stated that sIL-6R in co-culture of leukemia cells with bone-marrow stromal cells from WT or IL-6 246 KO mice could not be detected, this is a significant leap to then state that "the therapeutic benefit we see in vivo is independent of sIL-6R/IL-6 complexes binding to gp130." Also IL-6TS is influenced by both sIL-6R and gp130 levels, the latter of which is not measured. The authors should specifically target IL-6TS (which can be readily done with specific inhibitors) to confirm this key mode of signaling is (or isn't) playing a role.

Response to reviewers

We are sincerely appreciative of the comments made by the reviewers and the time taken to review this work. We have endeavored to address all of the issues raised in the second revision. This includes an extensive analysis of downstream IL-6 signaling, as well as a completely new RNA-Seq data set. The combined new data provided in this revision validate and extend our previous findings that IL6 deficiency “rewires” the tumor stroma (as opposed to the tumor cells, themselves) to promote chemotherapy-induced immune responses. We believe that this work has been substantially improved by the review process and the involvement of these reviewers.

Reviewer #1 (Remarks to the Author):

The Authors have convincingly addressed most of my comments on the initial submission, and the manuscript is now significantly improved. I have no additional comments.

We thank this reviewer for its extensive and thoughtful comments. Indeed, we believe that this work is substantially improved due to these insightful reviews.

Reviewer #2 (Remarks to the Author):

The authors showed that doxorubicin treatments can recruit immune cells to bone marrow, and that IL6 secreted by leukemic and stromal cells induces immune suppression in bone marrow microenvironment. Furthermore, IL6 depletion in bone marrow tumor microenvironment resulted in long-term T-cell mediated anti-cancer immune response. Finally, their study suggested IL6 depletion can synergize PD1-PDL1 immunotherapy. Although the findings of the DNA damage chemotherapy induced immune cell recruitment in TME has some novel aspect, overall the results shown in the manuscript lack innovation.

While the authors attempted to address concerns raised by the three previous reviewers and improve the scientific rigor, there are still a number of concerns.

1. The detailed mechanism of IL6 suppressed T-cell immune response in this study is not clear. They ruled out STAT3 activation in bone marrow tumor microenvironment in the B-ALL cancer model. However, IL6/IL6R/STAT3 signaling has been shown to be very important in pancreatic stromal cells (stellate cells) and PDAC tumor microenvironment. Since the authors used PDAC mouse model as their second animal model in the revised manuscript, it is unclear whether IL6-mediated cytotoxic T-cell suppression is STAT3-independent in PDAC TME.

We thank the reviewer for this suggestion. We now present data from bulk PDAC tumor samples showing that p-STAT3 levels do not change in the absence of IL-6 or following treatment. These data suggest that IL-6 suppressive effects are independent of STAT3 signaling in this solid tumor model as well.

Notably, this does not mean that IL-6 signaling through STAT3 is irrelevant to PDAC biology. Indeed, we have seen in other models that IL-6 signaling through STAT3 is critically chemoprotective. However, it does not seem to be the overarching mechanism in this context. We have tempered our text to not preclude some role for STAT3 signaling in this system.

2. The results from RNAseq are still underdeveloped, and the data they showed in the manuscript are very confusing. For example, the gene signatures in stimulatory and inhibitory T cell function gene sets both went down in IL6-KO cells in supplemental Figure 3 (or only down-regulated genes are listed?). Since the authors claimed a global difference in immune cells in TME between WT and IL6-KO mice, the ranking of all enriched gene sets, statistical significance and the enrichment scores (including FDR) from GSEA analysis should be included in the manuscript.

We appreciate the reviewers concern here. To address this issue, we have repeated our RNA-Seq experiments and now provide a completely revised data set – one that has sufficient high coverage to provide the detailed analysis requested by the reviewer. Notably, these new data confirms that IL-6 KO stroma (as opposed to less pronounced changes in tumor cells from IL-6 KO mice) show an enhanced inflammatory signature relative to WT stroma.

B-ALL and stromal cells were sorted from the bone marrow of WT and IL-6 KO mice, and RNA-sequencing was performed. DESeq2 was used to identify differentially expressed genes in the tumor and stroma of IL-6 KO mice relative to wild-type. GSEA analysis of the pre-ranked list using the cancer ‘Hallmarks’ collection from MSigDB identified the gain of inflammatory responses in IL-6 KO samples, suggesting a global difference in immune states between WT and IL-6 KO mice. The directionality of gene expression changes in these samples indicated that IL-6 KO leukemic mice have an enhanced immune response. This prompted us to perform Euclidean distance analysis to identify if these inflammatory responses are more prominent in tumor or stroma samples. We found the most variance pertained to the stroma samples, for both global normalized gene expression and, more importantly, for the genesets within the GSEA Hallmarks collection (Supplemental Figure 4).

Again, we are appreciative of this reviewer’s request to provide more extensive analysis of expression data, as this analysis has revealed two important results. First, rather than IL-6 loss having direct effects on tumor cells, IL-6 deficiency alters the stroma to create a permissive immune microenvironment. Second, the loss of IL-6 leads to an elevated immune signature in tumor stroma, including increased expression of genes that are canonical targets of IL-6 signaling, a possible compensatory response that may poise the environment for more robust anti-cancer immunity. These data provide mechanistic support for the major conclusion of this work that loss of a pro-inflammatory cytokine can lead to elevated anti-tumor immunity.

3. Variable numbers (and sometimes very low numbers) of mice were used in the different groups in the same experiment, which decreases the scientific rigor and statistical power.

Mouse numbers are indeed variable, but this occurs frequently when comparing diverse mouse cohorts. We respectfully argue that the use of distinct mouse numbers does not preclude statistical comparison of survival data – and have consulted our bioinformatics staff who support this idea. Based on rigorous statistical tests, our data reach statistical significance.

4. For PDAC retransplant, the tumor sizes in both groups may be too small (below 40 mm³) to interpret the results. No methodology was mentioned how the tumor size was measured.

We used electronic calipers to measure the tumor sizes and the methods section has been updated to clarify this. Tumors were at a size you could inspect by the naked eye and feel with your fingers, mice in the re-transplant experiment were still bearing the initially treated tumors on the opposite flank and needed to be sacrificed due growth in size of the initial tumor.

Reviewer #3 (Remarks to the Author):

The authors have addressed numerous concerns I raised previously, which have no doubt improved the quality of the work.

However, the authors have not addressed a key comment regarding mechanism, which is based on the statement "while stromal p-STAT3 levels increase in response to DOX treatment, there is no difference between WT and IL-6 KO mice (Fig. 2f), suggesting that the effects of IL-6 may occur independently of changes in STAT3 signaling". As requested before, authors should demonstrate what other well-documented signaling pathways downstream of IL-6 are playing a role by performing immunoblots/flow cytometry and using pathway-specific inhibitors.

We thank this reviewer for their positive comments regarding this work and their very helpful suggestions.

To further understand the molecular mechanism that mediates treatment resistance by IL-6, we performed immunoblot analysis of various IL-6 effectors on bone marrow lysates from B-ALL bearing mice (Supplemental Figure 3). Activation of S6 kinase (S6K), a target for PI3K/mTOR signaling, was not significantly changed by the absence of IL-6 nor exposure to DOX treatment. Similarly, activation of ERK1/2, a target for Ras/MAPK signaling, remained unchanged regardless of treatment conditions. We also assessed expression of the PI3K/mTOR and MEK/ERK/Ras pathways in B-ALL in our RNA-Seq data set and found no differences in expression in the absence of IL-6.

It is, however, possible that IL-6/STAT3/S6K signaling effects are occurring for short periods of times and/or at different time points than those scrutinized. In the revised manuscript, we temper our statement regarding the role of IL-6 – STAT3 signaling on this biology to allow for this possibility.

Experiments targeting STAT signaling with small molecules are potentially quite interesting, however Jak/Stat inhibitors change the biology of these B-cell malignancies – including potentially promoting accelerated progression and CNS dissemination of disease – in a manner that would preclude drawing definitive conclusions from these experiments.

Also, even though they stated that sIL-6R in co-culture of leukemia cells with bone-marrow stromal cells from WT or IL-6 KO mice could not be detected, this is a significant leap to then state that "the therapeutic benefit we see in vivo is independent of sIL-6R/IL-6 complexes binding to gp130." Also IL-6TS is influenced by both sIL-6R and sgp130 levels, the latter of which is not measured. The authors should specifically target IL-6TS (which can be readily done with specific inhibitors) to confirm this key mode of signaling is (or isn't) playing a role.

We thank the reviewer for this suggestion and attempted a limited version of this experiment. Our preliminary data suggests that blockade of IL-6-TS does not recapitulate the survival effects of global IL-6 blockade in the TME. Interestingly, IL-6-TS blockade may even prevent doxorubicin efficacy! Specifically, at a time point at which control mice are still alive following treatment, mice treated with IL-6-TS blockade are relapsing (see figure below).

We believe that this result is quite interesting (and thank the reviewer for the suggestion), but it is still preliminary, and we feel that we would need more mice, controls and additional data to definitively establish that relapse is accelerated in the presence of IL6-TS blockade. We show this data to the reviewer to suggest that IL6-TS blockade does not recapitulate the effects of IL6 deficiency, but believe that further investigation of this issue may be – given the lengthy nature of this review process – out of the scope of this current work.

Of note, our revised expression data shows that IL-6 deficiency leads to an “immune activated” stromal signature. Thus, IL-6 deficiency may paradoxically result in a context in which IL-6-TS mediates activation of stromal cells via additional (non-IL-6 ligands). Again, clarification of this data will require considerable additional study.

Reviewers' Comments:

Reviewer #2:

Remarks to the Author:

The authors have addressed all the main concerns from the reviewer.

Reviewer #3:

Remarks to the Author:

I appreciate the fact that the authors have attempted to address my comments to the best of their ability. While the manuscript is improved somewhat, there are remaining substantial mechanistic questions that remain unanswered regarding the signaling pathways involved and the mode of IL-6 signaling.

Response to reviewers

Reviewer #2 (Remarks to the Author):

The authors have addressed all the main concerns from the reviewer.

We thank the reviewer for their insightful and valuable comments.

Reviewer #3 (Remarks to the Author):

I appreciate the fact that the authors have attempted to address my comments to the best of their ability. While the manuscript is improved somewhat, there are remaining substantial mechanistic questions that remain unanswered regarding the signaling pathways involved and the mode of IL-6 signaling.

We thank the reviewer for their comments and suggestions.

We have added content to the discussion to highlight the remaining uncertainties regarding the mode of IL6 signaling and the pathways involved. Indeed, we are committed to bringing resolution to these issues in future work and believe publishing this work will engage the community to further interrogate the mechanisms underlying this significant clinical response.